# Male vs. Female Differences in Responding to Oxygen–Ozone Autohemotherapy (O_2_-O_3_-AHT) in Patients with Myalgic Encephalomyelitis/Chronic Fatigue Syndrome (ME/CFS)

**DOI:** 10.3390/jcm11010173

**Published:** 2021-12-29

**Authors:** Salvatore Chirumbolo, Luigi Valdenassi, Marianno Franzini, Sergio Pandolfi, Giovanni Ricevuti, Umberto Tirelli

**Affiliations:** 1Department of Neurosciences, Biomedicine and Movement Sciences, University of Verona, 37134 Verona, Italy; 2Italian Society of Oxygen Ozone Therapy (SIOOT), University of Pavia, 27100 Pavia, Italy; luigi.valdenassi@unipv.it (L.V.); marianno.franzini@gmail.com (M.F.); sergiopandolfis2@gmail.com (S.P.); giovanni.ricevuti@unipv.it (G.R.); 3Villa Mafalda Clinics, Via Monte delle Gioie 5 Rome, 00199 Roma, Italy; 4School of Pharmacy, Department of Drug Sciences, University of Pavia, 27100 Pavia, Italy; 5Tirelli Medical Group, 33170 Pordenone, Italy; utirelli@cro.it

**Keywords:** ME/CFS, ozone, oxygen–ozone therapy, fatigue, clinical trial

## Abstract

(1) Background: Myalgic Encephalomyelitis/Chronic Fatigue Syndrome (ME/CFS) is a syndrome that has fatigue as its major symptom. Evidence suggests that ozone is able to relieve ME/CFS-related fatigue in affected patients. (2) Objective: To evaluate whether differences exist between males and females in ozone therapy outputs in ME/CFS. (3) Methods: In total, 200 patients previously diagnosed with ME/CFS (mean age 33 ± 13 SD years) underwent treatment with oxygen–ozone autohemotherapy (O_2_-O_3_-AHT). Fatigue was investigated via an FSS 7-scoring questionnaire before and following 1 month after treatment. (4) Results: The Mann-Whitney test (MW test) assessed the significance of this difference (H = 13.8041, *p* = 0.0002), and female patients showed better outcomes than males. This difference was particularly striking in the youngest age cohort (14–29 years), and a KW test resulted in H = 7.1609, *p* = 0.007 for the Δ = 28.3% (males = 3.8, females = 5.3). (5) Conclusions: When treated with O_2_-O_3_-AHT, females respond better than males.

## 1. The Myalgic Encephalomyelitis/Chronic Fatigue Syndrome (ME/CFS) Challenge

Myalgic encephalomyelitis/chronic fatigue syndrome (ME/CFS) is a complex pathology, and was recently reviewed in [1]. ME/CFS must be considered a serious and long-term syndrome, which is characterized by fatigue and debilitating muscular–skeletal pain, conditions that affect many fundamental aspects of people’s social habits [2,3,4]. Fatigue is a major symptom in ME/CFS and its treatment is accounted for in many forerunners in Italy [5,6,7,8,9,10,11,12,13,14,15,16]. Tirelli et al. performed a study with 82 CFS patients living in northern Italy, showing that early symptoms occur between 24 and 40 years and that ME/CFS is primarily (3:1) observed in female subjects [15]. However, ME/CFS diagnosis is particularly burdensome, as patients with fatigue and other clinical signs are more often misdiagnosed with other chronic illnesses [17,18]; this is despite official diagnostic criteria for ME/CFS from the Centers for Disease Control and Prevention, the so-called IOM 2015 Diagnostic Criteria, having been updated in the CDC’s 1994 guidelines, which can be consulted elsewhere [19]. Fatigue is the leading symptom of ME/CFS, alongside other physical symptoms, such as headaches, tender lymph nodes, sore throat, poor sleep, poor concentration, reduced attention or memory, post-exertional malaise, muscular–skeletal pain and polyarthralgia, [20,21,22,23]. The diagnosis of ME/CFS therefore almost entirely based on fatigue-related symptoms [24,25].

The burdensome task to achieve a proper and sound diagnosis affects the therapeutic approach [26], despite some commendable attempts [5,7,11]; however, ME/CFS remains a considerable concern for clinics. Promising attempts in treating ME/CFS fatigue with oxygen–ozone autohemotherapy (O_2_-O_3_-AHT) were successfully performed by our group and others [27,28,29]. O_2_-O_3_-AHT may affect many complex issues in terms of immunity, most of which characterize the pathogenetic mechanisms causing ME/CFS [30,31]. Interestingly, ME/CFS might also have an oxidative stress causative pathogenesis [32,33]. This evidence, linked to the increasing awareness that ozone is able to regulate inflammation by targeting the oxidative stress signaling [32], thus suggesting several encouraging pieces of evidence for ME/CFS diagnoses [27], compelled us to treat fatigue in patients with O_2_-O_3_-AHT and investigate whether therapy outputs showed differences between male and female patients. 

## 2. Materials and Methods

### 2.1. Patient Recruitment

In total, 224 outpatients (mean age from the clinical centers of Pordenone and Gorle (Bergamo) were enrolled, having met the eligibility criteria agreed for the present study. Male mean age was 32.04 ± 18.45 SD and female mean age was 29.65 ± 11.54 SD (*p* = 0.354, in a Wilcoxon test). Of these patients, 200 entered the study; 19 escaped the study design because they referred to other therapy centers and were excluded. Five were formally accepted but never started for family and private reasons. Mean age was 33.08 ± 13.50 SD years [CI_95_ = 31.20–34.97], and median age was 33.14 years, comprising 69 men (34.5%) and 131 women. All patients were made aware of the therapy protocol and the use of the data for research purposes, according the recommendations of the Declaration of Helsinki.

### 2.2. Inclusion and Exclusion Criteria

Inclusion criteria were represented by outpatients referred to our clinical healthcare who were previously diagnosed with ME/CFS [23,34] and suffering from fatigue. These patients accepted and signed the informed consent for therapy and allowed for their data to be shared for research purposes. Exclusion criteria were represented by patients without ME/CFS, with other chronic and inflammation diseases, such as tumors or other immunological disorders and those who had taken pharmaceutical drugs in the previous 72 h; other exclusion criteria were those with chronic inflammatory and immune ailments such as autoimmunity, cancer or chronic inflammatory illness and pregnancy. 

### 2.3. Sample Size

Sample size was calculated to achieve an error range of about 10%. Referring to a population proportion of 51%, forecast data resulted in a 13.86% error with 50 patients, whereas there was an error of 9.80% (<10%) with 103 patients; therefore, 200 patients were within the minimal sample size with *p* < 0.001. The Cohen d statistics for the two independent groups, i.e., before O_2_-O_3_-AHT and following O_2_-O_3_-AHT, were successfully implemented (*p* = 0.004323 or Hedges’ *g* value). Moreover, Glass’ delta was *p* = 0.012444 (*p* < 0.02).

### 2.4. Patient Evaluation of Fatigue Symptomatology

An anamnestic interview and complete visitation of about 20–30 min were performed. Fatigue was the major symptom evaluated in the study as it was able to highlight patients’ overall clinical status in the most sound and reliable manner; this is due to its optimal performance features, stability over time and scant possibility of being overshadowed by other minor symptoms. Each patient was asked to respond to a 7-point scoring system, the Fatigue Severity Scale (FSS), before undergoing therapy and one month following therapy [35,36]. Results were collected as scores and statistically evaluated.

### 2.5. Patients’ Treatment with Oxygen–Ozone Autohemotherapy (O_2_-O_3_-AHT)

Patients underwent no fewer than two weekly sessions of major oxygen–ozone autohemotherapy, according to the protocol previously assessed by the Italian Society of Oxygen–Ozone Therapy (SIOOT) [27]. Briefly speaking, each patient underwent a treatment option requiring an ozone generator, compressed oxygen as a medical grade, a venipuncture syringe and a certified bag with an intravenous cannula for ozone therapy via autohemotherapy. A maximal volume of 200 mL of blood was usually withdrawn from each patient and collected in a CE-certified SANO_3_ bag, with automatic gentle mixing then immediately treated with 45 μg/mL of an O_3_ mixture in O_2_ (Multioxygen Medical 95 CPS, Gorle, Italy). This was finally reintroduced into the circulatory blood directly and within a few minutes [27,37]. Patients were followed up after 30 days following the second O_2_-O_3_-AHT session, and were asked to complete the FSS questionnaire, as previously agreed.

### 2.6. Statistics 

Data were collected and expressed as mean ± standard deviation, for quantitative values. Sample size was evaluated by assessing data and forecasting evaluations with Cohen d statistics and a Glass’ delta. Statistical inference, if any, was evaluated following non-parametric tests. Scores were evaluated by a Mann–Whitney test for two independent groups, with *p* < 0.05. Data were elaborated with SPSS v 24 software and Stata software for graphs.

## 3. Results

Figure 1 shows the difference (DELTA) in FSS score between female patients and male patients undergoing O_2_-O_3_-AHT. The average score was 5.14 ± 1.18 SD for females (CI_95_ = 4.865–5.4209) and 4.03 ± 1.80 SD for males (CI_95_ = 3.606–4.450) (mode: 6 and 5, respectively, median 5.5 and 5.0, respectively). The Mann–Whitney (MW) test assessed the significance at *p* < 0.05 of this difference (*p* = 0.0001), and female patients showed a better outcome than males. This difference was particularly striking in the youngest age cohort (14–29 years), as the MW test resulted in a *p* = 0.006 for the Δ = 28.3% (males = 3.8, females = 5.3). By arranging age groups into three clusters, i.e., 14–29 years, 30–49 years and ≥50, no difference in FSS score was reported for either females or males. Due to it having the highest heterogeneity in FSS response, the male group (see Figure 1), i.e., the MW test in the male age clusters, lacked statistical significance (*p* = 0.76). The O_2_-O_3_-AHT works optimally in an independent way with respect to elderly subjects of those of different ages. 

## 4. Discussion

Our results suggest that O_2_-O_3_-AHT is able to relieve fatigue in almost half of the whole cohort of ME/CFS patients. Female patients showed a higher ability, particularly during youth (14–29 years), to respond to O_2_-O_3_-AHT than males. Age clusters did not significantly affect the influence of O_2_-O_3_-AHT on both sexes. Thus far, no sound explanation can be attained to explain why female patients show better outcomes with O_2_-O_3_-AHT. This fact might be explained by the different endocrine endowment in males compared to females, at least in terms of ER receptors and their effect on T cell activation and NK cell functional activity [11]. The modulation of immunity may be a sound solution to counter ME/CFS fatigue, and ozone may be a possible approach [5,6,27]. We are therefore unable to fully elucidate how ozone can restore wellness in patients suffering from ME/CFS-related fatigue [27]. Cell biology should suggest that, in ME/CFS mitochondria, activity is greatly disturbed, generating an impairment in fundamental mitochondria-related activities, such as ROS signaling [38], leading to inflammation disorders [39,40]. During ME/CFS, an increase in CD4^+^CD25^+^Foxp3^+^ T regulatory (Treg) cells occurs [41], a circumstance that may be modulated by ozone [42]. ME/CFS pathogenesis involves the impairment of Th17 cells. The CCR6^+^ Th17 cells in ME/CFS secrete less IL-17 with respect to healthy subjects; moreover, their cell frequency in blood is lower and ozone can restore their numbers [43,44]. The immune micro-environment in ME/CFS is therefore fundamental for proper therapy [9,10,11]. Furthermore, 4-hydroxynonenal (4-HNE) induces the thioredoxin reductase 1 via Nrf2 signaling, and then increases the level of Tregs [45,46]. The ability of ozone to modulate immunity and inflammation via the Nrf2 system is particularly well noted [32]. Ozone may regulate nitric oxide (NO) and eNOS [32]. That said, ME/CFS patients should have normal NO alongside normal IL-6 levels, both before and after physical exercise upon fatigue symptoms, but should also show high levels of F2-isoprostanes, i.e., oxidative stress biomarkers, which are probably quenched by the activity of ozone on the Nrf2/Keap1/ARE system [47,48,49]. 

The ability of O_2_-O_3_-AHT to elicit a greater response in females than males may involve ER-beta signaling on T reg biology [50], and may signal the possible involvement of Nrf2 signaling, as elicited by O_2_-O_3_-AHT. This represents another issue that should be investigated in future studies [51,52]. 

## 5. Conclusions

Patients suffering from ME/CFS fatigue and being treated with O_2_-O_3_-AHT experience rapid relief of their symptoms. Female subjects are able to respond to O_2_-O_3_-AHT and reduce fatigue symptoms better than males. Further insights are needed to elucidate the mechanism by which these differences occur. 

## Figures and Tables

**Figure 1 jcm-11-00173-f001:**
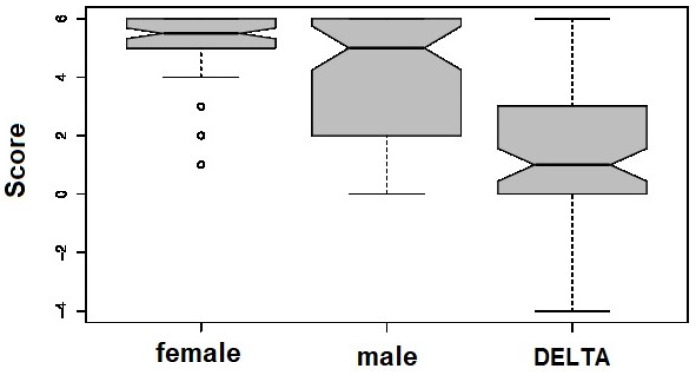
Box plot representation of the FSS differential response between male and female patients with ME/CFS treated with O_2_-O_3_-AHT.

## Data Availability

Data repository can be requested to the Prof Umberto Tirelli, at your disposal and to Prof M Franzini at SIOOT (Bergamo).

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
