# Peer review of "Male vs. Female Differences in Responding to Oxygen–Ozone Autohemotherapy (O_2_-O_3_-AHT) in Patients with Myalgic Encephalomyelitis/Chronic Fatigue Syndrome (ME/CFS)"

_jcm, 2021, doi:10.3390/jcm11010173_

Round 1

Reviewer 1 Report

The manuscript is well written and has important information about a possible treatment option for fatigue in patients with ME/CFS. It is not a placebo-controlled study and this should be mentioned in discussion. It is really intrigating why there is such a difference between female and male groups.

There are some minor points:

Line 30: it is possible to understand what authors mean with "their", but strictly, whose "lives"aspects may abruptly shut down?

Line 60: I miss the reason  for the 19 escapes and the 5 that never started.

Line 62: the sentense "the majority of patients were female subjects" is superfluous.

Author Response

The manuscript is well written and has important information about a possible treatment option for fatigue in patients with ME/CFS. It is not a placebo-controlled study and this should be mentioned in discussion. It is really intrigating why there is such a difference between female and male groups.

Authors’ rebuttal: Many thanks for this consideration

There are some minor points:

Line 30: it is possible to understand what authors mean with "their", but strictly, whose "lives"aspects may abruptly shut down?

Authors’ rebuttal: The sentence was revised as follows: ME/CFS must be considered a serious and long-term syndrome, which is characterized by fatigue and debilitating muscular-skeletal pain, conditions that may abruptly affect many fundamental aspects of people’s social habits

Line 60: I miss the reason  for the 19 escapes and the 5 that never started.

Authors’ rebuttal: this part was revised accordingly

Line 62: the sentense "the majority of patients were female subjects" is superfluous.

Authors’ rebuttal: Removed, accordingly

Reviewer 2 Report

Thank you for taking the opportunity to review this article.

This work represents an important area of inquiry that is relevant to the readership of this journal. However, additional details are needed for optimal review.

  1. Abstract: The objectives of the work should be added (line 16).
  2. Line 54: please replace “femal” with “female”.
  3. Line 55: delete that phrase.
  4. Section 2.1 .: Report the mean ages of women and men, as well as the p-value of the test.
  5. Section 2.3 .: Replace “test” with “statistics” and “separate” with “independent” (line 79).
  6. Section 2.6 .: Replace “test” with “statistics” (line 105). The Kruskall-Wallis test is suitable for 3 or more groups, in this case, with only 2 groups, you should use the Mann-Whitney (M-W) test.
  7. Results: Report the median of each group, as well as its 95% confidence interval and use the M-W test.

Author Response

Thank you for taking the opportunity to review this article.

This work represents an important area of inquiry that is relevant to the readership of this journal. However, additional details are needed for optimal review.

  1. Abstract: The objectives of the work should be added (line 16).

Authors’ rebuttal: added, accordingly

  1. Line 54: please replace “femal” with “female”.

Authors’ rebuttal: Done

  1. Line 55: delete that phrase.

Authors’ rebuttal: Revised

  1. Section 2.1 .: Report the mean ages of women and men, as well as the p-value of the test.

Authors rebuttal: Done

  1. Section 2.3 .: Replace “test” with “statistics” and “separate” with “independent” (line 79).

Authors’ rebuttal: Done

  1. Section 2.6 .: Replace “test” with “statistics” (line 105). The Kruskall-Wallis test is suitable for 3 or more groups, in this case, with only 2 groups, you should use the Mann-Whitney (M-W) test.

Authors’ rebuttal. Done, accordingly. Typically, a Kruskal-Wallis H test is used when you have three or more categorical, independent groups, but it can be used for just two groups, i.e., that a Mann-Whitney U test is more commonly used for two groups but not mandatory. Anyway, we adsopted a Mann Whitney test accordingly

  1. Results: Report the median of each group, as well as its 95% confidence interval and use the M-W test.

Authors’ rebuttal: Done